# Metagenomic Analysis Reveals High Abundance of Torque Teno Mini Virus in the Respiratory Tract of Children with Acute Respiratory Illness

**DOI:** 10.3390/v14050955

**Published:** 2022-05-03

**Authors:** Antonin Bal, Gregory Destras, Marina Sabatier, Maxime Pichon, Hadrien Regue, Guy Oriol, Yves Gillet, Bruno Lina, Karen Brengel-Pesce, Laurence Josset, Florence Morfin

**Affiliations:** 1Laboratoire de Virologie, Institut des Agents Infectieux, Laboratoire Associé au Centre National de Référence des Virus des Infections Respiratoires, Hospices Civils de Lyon, 69004 Lyon, France; antonin.bal@chu-lyon.fr (A.B.); gregory.destras@chu-lyon.fr (G.D.); marina.sabatier@hotmail.fr (M.S.); bruno.lina@chu-lyon.fr (B.L.); 2Univ Lyon, Université Lyon 1, CIRI, Inserm U1111 CNRS UMR5308, Virpath, 69007 Lyon, France; 3GenEPII Platform, Institut des Agents Infectieux, Hospices Civils de Lyon, 69004 Lyon, France; hadrien.regue@chu-lyon.fr; 4Bacteriology Laboratory, Infectious Agents Department, Centre Hospitalier Universitaire de Poitiers, 86021 Poitiers, France; maxime.pichon@chu-poitiers.fr; 5Inserm U1070 Pharmacology of Antimicrobial Agents and Resistance, University of Poitiers, 86073 Poitiers, France; 6Laboratoire Commun de Recherche HCL-bioMerieux, Centre Hospitalier Lyon Sud, 69495 Pierre-Bénite, France; guy.oriol@biomerieux.com (G.O.); karen.brengel-pesce@biomerieux.com (K.B.-P.); 7Hospices Civils de Lyon, Urgences Pédiatriques, Hôpital Femme Mère Enfant, 69500 Bron, France; yves.gillet@chu-lyon.fr

**Keywords:** torque teno virus, torque teno mini virus, *Anelloviridae*, metagenomics, acute respiratory infection, respiratory virus

## Abstract

Human *Anelloviridae* is a highly prevalent viral family, including three main genera—*Alphatorquevirus* (Torque teno virus, TTV), *Betatorquevirus* (Torque teno mini virus, TTMV), and *Gammatorquevirus* (Torque teno midi virus, TTMDV). To date, the characterization of *Anelloviridae* in the respiratory tract of children with acute respiratory infection (ARI) has been poorly reported and mainly focused on TTV. We performed a metagenomic analysis of eight respiratory samples collected from children with an ARI of unknown etiology (eight samples tested negative with a multiplex PCR assay, out of the 39 samples initially selected based on negative routine diagnostic testing). A total of 19 pediatric respiratory samples that tested positive for respiratory syncytial virus (RSV, *n* = 13) or influenza virus (*n* = 6) were also sequenced. *Anelloviridae* reads were detected in 16/27 samples, including 6/8 negative samples, 7/13 RSV samples and 3/6 influenza samples. For samples with a detection of at least one *Anelloviridae* genus, TTMV represented 87.1 (66.1–99.2)% of *Anelloviridae* reads, while TTV and TTMDV represented 0.8 (0.0–9.6)% and 0.7 (0.0–7.1)%, respectively (*p* < 0.001). Our findings highlight a high prevalence of TTMV in respiratory samples of children with an ARI of unknown etiology, as well as in samples with an RSV or influenza infection. Larger studies are needed to explore the role of TTMV in childhood respiratory diseases.

## 1. Introduction

Human *Anelloviridae* is a highly prevalent viral family characterized by an important genetic diversity. More than eighty species are grouped into three main genera: *Alphatorquevirus* (Torque teno virus, TTV), *Betatorquevirus* (Torque teno mini virus, TTMV) and *Gammatorquevirus* (Torque teno midi virus, TTMDV) [1,2]. The viral load of TTV in plasma or blood is considered as a surrogate marker of immune competence and might be used to assess the immune status of transplant patients [3,4]. TTMV and TTMDV have been less studied, but the prevalence in blood is considerably lower than that of TTV [5,6]. *Anelloviridae* has also been detected in almost all parts of the human body, including the respiratory tract, and is recognized as the main component of the human viral flora [5,7,8]. While considered to be non-pathogenic, *Anellovriridae* might be associated with the occurrence of some disorders, including respiratory disorders in childhood [9,10,11,12,13]. Co-infection between TTV and common respiratory viruses have been reported, but the detailed composition of *Anelloviridae,* including the characterization of TTMV and TTMDV, has been poorly explored in children with respiratory virus infections [14,15].

Due to a complex genetic diversity, PCR-based detection of *Anelloviridae* genera might be difficult to optimize, and no diagnostic PCR has been validated in respiratory samples so far [5,8,16]. As untargeted methods, metagenomic next-generation sequencing (mNGS) has become a powerful tool for the characterization of the whole viral communities in clinical samples, and has been used to describe the abundance and/or dynamics of TTV, TTMV and TTMDV [4,6,17,18]. Herein, we aim to describe the *Anelloviridae* composition within the respiratory virome of children under five years presenting an ARI with or without identified etiologic agents.

## 2. Materials and Methods

### 2.1. Samples Selection

A retrospective study was conducted on samples received at the virology laboratory at the University Hospital of Lyon, France, between 2010 and 2016. Nasopharyngeal aspirates from hospitalized children under five years old with ARI were selected according to the following criteria: (1) absence of documented infections on two consecutive samples (collected up to two weeks apart) with routine techniques that included bacterial/viral cultures, reverse-transcriptase polymerase chain reaction (RT-PCR) assay detecting human rhinovirus, and respiratory syncytial (RSV) and influenza viruses (MWS r-gene™ respiratory panel; bioMérieux, Lyon, France); (2) negative results with a large screening multiplex PCR (FilmArray^®^ Respiratory Panel, bioMérieux, Lyon, France). This assay allows to detect the main viral and bacterial respiratory pathogens, including adenovirus, human coronavirus (229E, HKU1, OC43, NL63), human metapneumovirus, human rhinovirus/enterovirus, influenza (A, A/H1, A/H1-2009, A/H3, B), human parainfluenza virus, RSV, bordetella pertussis, chlamydophila pneumonia and mycoplasma pneumonia.

Nineteen respiratory samples from hospitalized children under five with ARI that tested positive with RSV RT-PCR (*n* = 13) or influenza RT-PCR (*n* = 6) were also sequenced.

### 2.2. Metagenomic Workflow

A metagenomic workflow including quality controls, evaluated for the detection of a comprehensive panel of DNA and RNA viruses in respiratory samples, was used [19]. Briefly, after sample viral enrichment, total nucleic acid was extracted, randomly amplified, and Illumina libraries were prepared using the Nextera XT DNA Library preparation kit, according to the manufacturer’s recommendations (Illumina, San Diego, CA, USA). Libraries were sequenced on Illumina NextSeq500™ platform with mid-output 2 × 150 bp flowcells. Raw fastq files generated were cleaned with cutadapt (version 1.18). The host reads were then removed using bwa mem (version 0.7.8) alignment to the human genome (GRCh37.p2). Unmapped reads extracted with samtools tool (version 1.3.1) were then mapped to the nr database (non-redundant protein sequences database) (downloaded on 25 September 2018) using Diamond (version 0.9.22). We also explored the prevalence of *Anelloviridae* species, as previously described [6]. Briefly, non-human NGS reads were aligned using BLAST on a manually curated database composed of 56 reference sequences of human *Anelloviridae* (TTV-1 to TTV-29, TTMV-1 to TTMV-12, TTMDV-1 and TTMDV-15).

To correct the difference in sequencing depth between samples, the number of reads were normalized in reads per million mapped reads (RPM). To reduce false positive results, only viruses with a count of >1 RPM were considered.

### 2.3. Statistical Analysis

Statistical analyses were conducted using R software, version 4.0.5 (R Foundation for Statistical Computing). Continuous variables are presented as the median with interquartile range (IQR) and compared using non-parametric Kruskal–Wallis or Mann–Whitney tests. Differences are considered significant at *p* < 0.05.

## 3. Results

### 3.1. Study Population

Between 2010 and 2016, 39 hospitalized patients with ARI were tested negative on two successive samples with routine diagnostic techniques. Among them, eight remained negative after testing with a large multiplex PCR assay and were analyzed in mNGS. The median (IQR) age was 12.0 (6.3–22.5) months; three patients had comorbidities, including spinal muscular atrophy, sickle cell anemia and cardiac congenital disease. Two patients presented severe respiratory distress that required admission in the intensive care unit (ICU).

Nineteen respiratory samples from hospitalized children with ARI that tested positive for RSV (*n* = 13) or influenza (*n* = 6) were sequenced. The median age was 2.3 (1.3–2.8) months and 1.3 (1.0–4.1) months for RSV and influenza patients, respectively; no ICU admission nor comorbidities were noticed.

### 3.2. Composition of Metagenomic Sequences

Libraries were sequenced to a median of 11,880,612 (9,085,799–15,865,360) reads, passing quality filters. Viral reads represented 0.7 (0.2–1.8)% of the total reads generated from samples that tested negative (vs. 1.2 (0.3–4.2)% and 0.3 (0.03–0.7)% for RSV and influenza samples, respectively, *p* = 0.63). Of note, viral reads mapping to RSV or influenza were detected in all samples of the RSV or influenza groups, respectively.

Viral contamination represented 0.04% of total reads detected in the no-template control and mainly (97.5%) derived from bacteriophages. General metagenomic data are summarized in Table 1.

### 3.3. Anelloviridae Abundance and Composition within the Respiratory Virome

Among the 27 patients included in the present study, *Anelloviridae* reads were detected in 16/27 (59.3%) samples, including 6/8 (75%) negative samples, 7/13 (53.8%) RSV samples and 3/6 (50%) influenza samples (Figure 1). The median abundances of *Anelloviridae* reads were 48.2 (24.6–6066.0) RPM for influenza samples, 51.4 (27.0–370.3) RPM for RSV samples and 1131.8 (181.6–2761.7) RPM for negative samples (*p* = 0.4).

To investigate the composition of *Anelloviridae*, relative abundances of TTMV, TTV and TTMDV were computed according to the normalized number of reads identified as genus level. For samples with a detection of specific reads mapping to at least one *Anelloviridae* genus (*n* = 13), TTMV represented 87.1 (66.1–99.2)% of *Anelloviridae* reads, while TTV and TTMDV represented 0.8 (0.0–9.6)% and 0.7 (0.0–7.1)%, respectively (*p* < 0.001; Figure 2). No significant differences were observed between the relative abundance of TTV vs. TTMDV (*p* = 0.57). Of note, for 2/13 samples, the main genus detected was not TTMV (TTMDV for one negative sample and TTV for one RSV sample, Figure 2). No significant differences regarding *Anelloviridae* genus abundances were noticed between negative, RSV and influenza samples.

At the species level, a high inter-individual variability was noticed with multiple species detection for most individuals. TTMV-10 and TTMV-5 were the two most frequent species detected in the negative group, while the most prevalent species in the RSV group were TTMV-10, followed by TTMV-11, TTMV-6, and TTMV-5. Regarding the two patients with a detection of *Anelloviridae* species in the influenza group, TTMV-10, TTMV-5, TTMV-4, TTMV-3 and TTMV-1 were co-detected (Figure 3).

## 4. Discussion

In this present study, *Anelloviridae* was detected in about 60% of respiratory samples of children under five, highlighting a high inter-individual variability in *Anelloviridae* detection, as previously reported in blood or plasma samples [6,8,14]. Furthermore, we found a high abundance of TTMV in samples associated with or without viral pathogen detection (RSV or influenza). Although considered as commensal virome, *Anelloviridae* might contribute to respiratory diseases by activating the production of inflammatory cytokines [11,13]. TTMV has already been identified in several pathogenic settings, including empyema, encephalitis and periodontitis, suggesting a possible association of TTMV infection with inflammation [10,20,21]. In addition, in vitro replication of TTMV in alveolar epithelial cells has been demonstrated [10] and a recent mNGS study found that the upper respiratory virome composition of children with pneumonia was mainly represented by TTMV [12].

In contrast, Wang et al., showed that the majority of *Anelloviridae* reads mapped to TTV in children with severe ARI as well as in the control group constituted of 15 children without respiratory symptoms [9]. In addition, no differences in TTMV detection were noticed in respiratory samples of febrile pediatric patients compared to afebrile controls [22]. The differences in the *Anelloviridae* composition within the respiratory tract might be explained by clinical features, including the level of immunosuppression and the presence of viral co-infections, as well as by age, sex, geographical, or environmental factors which may influence the virome composition [23]. Importantly, we used the same mNGS methods as a previous study performed on the plasma samples of autologous stem cell transplant patients where TTV represented the most abundant genus (91.7% of *Anelloviridae* reads), while TTMDV and TTMV represented 6.4% and 1.9%, respectively [6].

As TTMV were found in all groups herein and in line with the limited number of patients included, we could not draw a conclusion regarding the respiratory pathogenicity of TTMV. This needs to be investigated in larger pediatric cohorts, including a healthy control group. The impact of TTMV levels, TTMV species diversity and TTMV co-infection (found in about 50% of influenza and RSV samples herein) on disease severity also need to be explored. Studies focusing on TTV found that a higher level or a co-infection with common respiratory viruses were associated with a more severe disease [14,24].

Overall, the present study emphasizes the high prevalence of *Anelloviridae* within the respiratory tract of children with ARI, and that the *Anelloviridae* composition is mainly represented by TTMV in the presence, or not, of common respiratory viruses. The characterization of whole viral communities is crucial for understanding the complex role of the human virome in respiratory diseases.

## Figures and Tables

**Figure 1 viruses-14-00955-f001:**
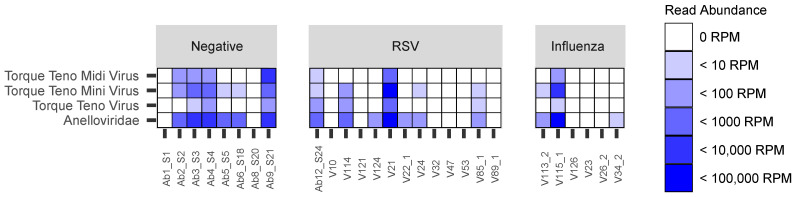
Abundance of *Anelloviridae*, TTV, TTMV and TTMDV in negative, RSV and influenza samples. *Anelloviridae* reads include TTV, TTMV, TTMDV reads, as well as reads from unclassified *Anelloviridae*. To correct the difference of sequencing depth between samples, the number of reads were normalized in reads per million mapped reads (RPM). Only viruses with a count of >1 RPM were considered.

**Figure 2 viruses-14-00955-f002:**
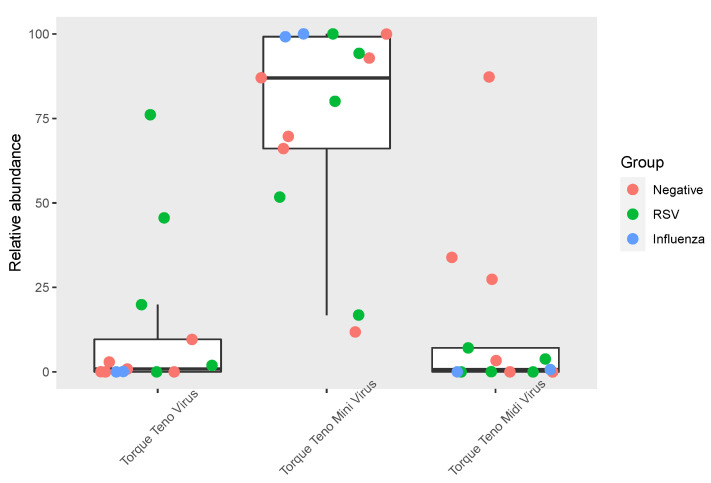
Relative abundance of *Anelloviridae* genera among negative, RSV and influenza samples. Relative abundances of TTMV, TTV and TTMDV were computed according to the normalized number of reads identified as genus level. The three groups (negative, RSV and influenza) are represented by dots of different colors.

**Figure 3 viruses-14-00955-f003:**
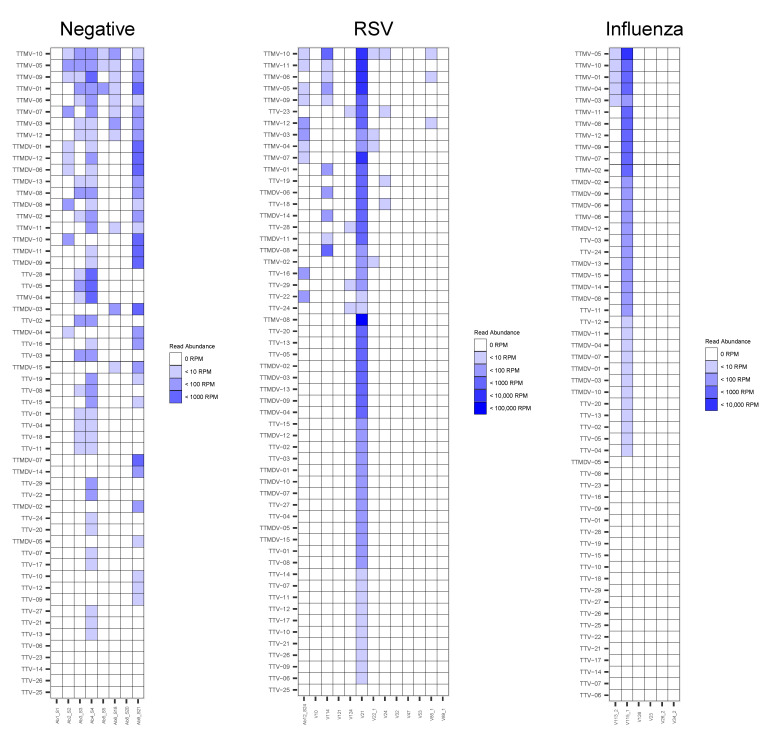
Abundance of *Anelloviridae* species in negative, RSV and Influenza samples. Species are ranked by frequency of detection in each group. To correct difference of sequencing depth between samples, number of reads were normalized in read per millon mapped read (RPM). Only viruses with a count >1 RPM were considered.

**Table 1 viruses-14-00955-t001:** Composition of metagenomic reads generated from samples that tested negative and samples that tested positive for RSV or influenza.

Group	Sample ID	Total Number of Reads	Number of Reads for PCR-Positive Virus	% Human	% Viruses	% Bacteria
Negative samples (*n* = 8)	Ab1_S1	13,242,872	NA	68.1	0.2	39.8
	Ab2_S2	21,124,748	NA	53.7	1.0	36.1
	Ab3_S3	18,613,428	NA	88.5	2.7	32.7
	Ab4_S4	5,729,468	NA	40.9	2.1	38.3
	Ab5_S5	6,506,898	NA	74.6	0.4	38.6
	Ab6_S18	9,719,338	NA	58.1	0.2	36.8
	Ab8_S20	12,453,484	NA	58.9	0.2	50.2
	Ab9_S21	5,124,554	NA	42.2	1.7	42.1
RSV samples (*n* = 13)	Ab12_S24	10,533,108	232,827	67.6	1.6	35.2
	V10	4,547,410	102	87.0	0.0	3.7
	V32	14,911,096	57,606	88.5	0.4	2.1
	V53	10,826,666	361,365	88.8	3.4	1.5
	V89_1	6,622,864	58,284	87.5	0.9	2.2
	V47	11,747,124	182,776	85.9	1.6	3.1
	V22_1	16,819,624	77	87.6	0.0	2.8
	V124	19,598,588	540,934	84.3	2.8	3.0
	V121	14,490,590	62,279	87.9	0.4	2.1
	V24	8,452,260	81,253	8.6	24.0	7.0
	V85_1	19,138,760	892	83.8	0.0	4.2
	V114	12,516,284	1,673,602	52.0	13.6	9.7
	V21	16,928,282	406,594	64.9	6.6	6.0
Influenza samples (*n* = 6)	V126	7,910,142	1125	87.2	0.01	3.6
	V23	18,131,472	103	90.7	0.001	0.6
	V26_2	13,672,726	53,556	82.7	0.5	4.8
	V34_2	11,880,612	8058	90.0	0.1	1.4
	V113_2	10,944,276	78,193	49.6	0.7	16.0
	V115_1	11,403,046	2108	3.0	59.9	6.7
NTC	NTC	7,210,744	NA	8.5	0.04	46.5

NTC: No-template control, RSV: respiratory syncytial virus, NA: not applicable.

## Data Availability

The data that support the findings of this study are available upon request from the authors.

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
