# Peer review of "Metagenomic Analysis Reveals High Abundance of Torque Teno Mini Virus in the Respiratory Tract of Children with Acute Respiratory Illness"

_viruses, 2022, doi:10.3390/v14050955_

Round 1

Reviewer 1 Report

The authors present a metagenomic analysis of pediatric respiratory samples, unknown etiology or tested previously positive for the respiratory syncytial virus, influenza virus. In general, the data is deemed valuable, and the analysis is technically sound. However, there is room for improvement.

Major concern:

The presentation of the analysis needs to be improved. For instance, a typical analysis of metagenomic data should involve the functional and taxonomic profiles respectively. The in-depth comparison, of whether there are similarities/differences in the sequenced viruses between each group is missing. What is the correlation to other available genomes in the GenBank from healthy/hospitalized patients? That’s the advantage of the NGS data otherwise, the NGS could be replaced with PCR.  

Minor concerns:

  1. Line 14: None of torque teno virus, torque teno mini virus and torque teno midi virus is classified as genus according to the ICTV. If authors want to mention main genera, then should use Alphatorquevirus Betatorquevirus, Gammatorquevirus.

  1. Description of which sequencing system and setup was used is missing. 

  1. Line 116: “Libraries were sequenced to a median of 11,880,612 (9,085,799-15,865,360) reads passing quality filters.” In table 1, there are 8 samples with a lower number of reads than the lowest number of sequenced reads is stated in the text. Please clarify.

Author Response

The authors present a metagenomic analysis of pediatric respiratory samples, unknown etiology or tested previously positive for the respiratory syncytial virus, influenza virus.

In general, the data is deemed valuable, and the analysis is technically sound. However, there is room for improvement.

Major concern:

Q1. The presentation of the analysis needs to be improved. For instance, a typical analysis of metagenomic data should involve the functional and taxonomic profiles respectively. The in-depth comparison, of whether there are similarities/differences in the sequenced viruses between each group is missing. What is the correlation to other available genomes in the GenBank from healthy/hospitalized patients? That’s the advantage of the NGS data otherwise, the NGS could be replaced with PCR. 

A1. We thank the reviewer for this comment.

In order to compare the prevalence of Anelloviridae species between the three groups, we performed additional analyses in implementing a dedicated bioinformatics pipeline for the detection of Anelloviridae species, as previously described (https://www.mdpi.com/1999-4915/10/11/633). Briefly, nonhuman NGS reads were aligned using BLAST on a manually curated database composed of 56 reference sequences of human Anelloviridae (TTV-1 to TTV-29, TTMV-1 to TTMV-12, TTMDV-1 and TTMDV-15). 

We found a high inter-individual variability with multiple species detection for most individuals. TTMV-10 and TTMV-5 were the two most frequent species detected in the negative group while the most prevalent species was TTMV-10 followed by TTMV-11, TTMV-6, and TTMV-5 in the RSV group. Regarding the two patients with a detection of Anelloviridae species in the Influenza group, TTMV-10, TTMV-5, TTMV-4, TTMV-3 and TTMV-1 were co-detected at the same level.

We added these new data in Methods and Result sections.

Methods:

“We explored the prevalence of Anelloviridae species as previously described (6). Briefly, nonhuman NGS reads were aligned using BLAST on a manually curated database composed of 56 reference sequences of human Anelloviridae (TTV-1 to TTV-29, TTMV-1 to TTMV-12, TTMDV-1 and TTMDV-15)”

Results:

“At species level, a high inter-individual variability was noticed with multiple species detection for most individuals. TTMV-10 and TTMV-5 were the two most frequent species detected in the negative group while the most prevalent species in the RSV group was TTMV-10 followed by TTMV-11, TTMV-6, and TTMV-5. Regarding the two patients with a detection of Anelloviridae species in the Influenza group, TTMV-10, TTMV-5, TTMV-4, TTMV-3 and TTMV-1 were co-detected at the same level”

We also made a new Figure to illustrate these results (Figure 3).

We did not investigate the Anelloviridae species composition according to the disease severity as all individuals of the present study were hospitalized but did not require ICU admission. Furthermore we did not include healthy patients. However we discussed this point in Discussion section.

“The impact of TTMV level, TTMV species diversity and TTMV co-infection (found in about 50% of Influenza and RSV samples herein) on disease severity also need to be explored.”

Minor concerns:

Q2. Line 14: None of torque teno virus, torque teno mini virus and torque teno midi virus is classified as genus according to the ICTV. If authors want to mention main genera, then should use Alphatorquevirus Betatorquevirus, Gammatorquevirus.

A2. The corrections have been made accordingly.

Q3. Description of which sequencing system and setup was used is missing.

A3.  Libraries were sequenced on Illumina NextSeq500™ platform with mid-output 2 × 150 bp flowcells. We added this point in Methods section.

Q4. Line 116: “Libraries were sequenced to a median of 11,880,612 (9,085,799-15,865,360) reads passing quality filters.” In table 1, there are 8 samples with a lower number of reads than the lowest number of sequenced reads is stated in the text. Please clarify.

A4.  The numbers in parentheses (9,085,799-15,865,360) indicate the interquartile range. We clarified this point in Methods section.

“Continuous variables are presented as median with interquartile range (IQR)”

Reviewer 2 Report

In the manuscript “Metagenomic Analysis Reveals High Abundance of Torque Teno mini virus (Human Anelloviridae) in the Respiratory Tract of Children with Acute Respiratory Illness”, the authors described a study using metagenomics NGS to detect the presence of three types of human Anellovirus in the nasopharyngeal aspirate specimens from children with or without other respiratory virus infection. The authors claim that the presence of anellovirus in the specimens was ubiquitous and TTMV was the most commonly seen anellovirus.

The understanding of human anellovirus and their impact on human diseases is still limited. This manuscript adds some value to the literature in this area.

Here are my comments and suggestions.

  1. Line 99. Please review the description of the consent process. The study participants were all young children. Consent from legal guardians should be obtained as well.

  1. The age of the negative viral infection group and the RSV and flu group were very different. I wonder if there was a correlation between the presence of anellovirus and the age of the participants as the older the children, the more likely they expose to the virus.

  1. Line 118-119. Are there any statistical differences?

  1. Table 1. Typo of ‘Nomber’. Some of the numbers used “,” instead of “.” for decimals.

  1. Participant V24 had 24% of viral reads and V114 had 13.6% viral reads. Did the authors find viruses other than RSV, Influenza or anellovirus in these participants?

  1. Line 145-146. Please also compare TTV vs. TTMDV.

Author Response

In the manuscript “Metagenomic Analysis Reveals High Abundance of Torque Teno mini virus (Human Anelloviridae) in the Respiratory Tract of Children with Acute Respiratory Illness”, the authors described a study using metagenomics NGS to detect the presence of three types of human Anellovirus in the nasopharyngeal aspirate specimens from children with or without other respiratory virus infection. The authors claim that the presence of anellovirus in the specimens was ubiquitous and TTMV was the most commonly seen anellovirus.

The understanding of human anellovirus and their impact on human diseases is still limited. This manuscript adds some value to the literature in this area.

Here are my comments and suggestions.

Q1. Line 99. Please review the description of the consent process. The study participants were all young children. Consent from legal guardians should be obtained as well.

A1. The samples used for this retrospective study were collected during routine patient management and no additional samples were used.  For this non interventional study, consent is not mandatory according to the French regulations. During their hospitalization in the HCL, legal guardians are made aware that de-identified data may be used for research purposes, and they can opt out if they object to the use of their data. This study was approved by the ethics committee of HCL as well as by national authorities (national data protection agency).

Q2. The age of the negative viral infection group and the RSV and flu group were very different. I wonder if there was a correlation between the presence of anellovirus and the age of the participants as the older the children, the more likely they expose to the virus.

A2. We thank the reviewer for this comment. We did not find a significant correlation between the age of participants and normalized read counts of Anelloviridae (r = 0.0514 CI 95% [-0.3352 ; 0.4231]). We agree that virome composition and abundance could vary according to the age of participant and underlined this point in the Discussion section of the revised manuscript.

“The differences in the Anelloviridae composition within the respiratory tract might be explained by clinical features including the level of immunosuppression, the presence of viral co-infections as well as by age, sex, geographical, or environmental factors which may influence the virome composition”

Q3. Line 118-119. Are there any statistical differences?

A3. No significant differences in were noted between the viral read percentages of the three groups (p=0.63 with Kruskal-Wallis test). We added the p-value in the Results section of the revised version of the manuscript.

Q4. Table 1. Typo of ‘Nomber’. Some of the numbers used “,” instead of “.” for decimals.

A4. The corrections have been made accordingly.

Q5. Participant V24 had 24% of viral reads and V114 had 13.6% viral reads. Did the authors find viruses other than RSV, Influenza or anellovirus in these participants?

A5. For these two RSV samples, RSV reads represented 99.1 and 4.0 % of viral reads for V114 and V24, respectively. For V24 sample, 95.1% of viral reads was represented by Herpesviridae reads.

Q6. Line 145-146. Please also compare TTV vs. TTMDV.

A6.  No significant differences were observed between the relative abundance of TTV vs TTMDV (p=0.57). We added this point in the Results section of the revised version of the manuscript.

 “For samples with a detection of specific reads mapping to at least one Anelloviridae genus (n=13), TTMV represented 87.1 (66.1-99.2) % of Anelloviridae reads while TTV and TTMDV represented 0.8 (0.0-9.6) % and 0.7 (0.0-7.1) %, respectively (p<0.001). No significant difference was observed between the relative abundance of TTV vs TTMDV (p=0.57)”

Round 2

Reviewer 1 Report

The authors have addressed all my comments. 

Reviewer 2 Report

The authors have addressed my comments.